# Guanylate cyclase 1 relies on rhodopsin for intracellular stability and ciliary trafficking

**Jillian N Pearring[1], William J Spencer[1,2], Eric C Lieu[1], Vadim Y Arshavsky[1,2]***

[1]Department of Ophthalmology, Duke University Medical Center, Durham, United States; [2]Department of Pharmacology and Cancer Biology, Duke University Medical Center, Durham, United States

**Abstract** Sensory cilia are populated by a select group of signaling proteins that detect environmental stimuli. How these molecules are delivered to the sensory cilium and whether they rely on one another for specific transport remains poorly understood. Here, we investigated whether the visual pigment, rhodopsin, is critical for delivering other signaling proteins to the sensory cilium of photoreceptor cells, the outer segment. Rhodopsin is the most abundant outer segment protein and its proper transport is essential for formation of this organelle, suggesting that such a dependency might exist. Indeed, we demonstrated that guanylate cyclase-1, producing the cGMP second messenger in photoreceptors, requires rhodopsin for intracellular stability and outer segment delivery. We elucidated this dependency by showing that guanylate cyclase-1 is a novel rhodopsin-binding protein. These findings expand rhodopsin's role in vision from being a visual pigment and major outer segment building block to directing trafficking of another key signaling protein.

**\*For correspondence:** vadim.
arshavsky@duke.edu

**Competing interests:** The authors declare that no competing interests exist.

## Introduction

Photoreceptor cells transform information entering the eye as photons into patterns of neuronal electrical activity. This transformation takes place in the sensory cilium organelle, the outer segment. Outer segments are built from a relatively small set of structural and signaling proteins, including components of the classical G protein-coupled receptor (GPCR) phototransduction cascade. Such a distinct functional and morphological specialization allow outer segments to serve as a nearly unmatched model system for studying general principles of GPCR signaling (*Arshavsky et al., 2002*) and, in more recent years, a model for ciliary trafficking (*Garcia-Gonzalo and Reiter, 2012*; *Nemet et al., 2015*; *Pearring et al., 2013*; *Schou et al., 2015*; *Wang and Deretic, 2014*).

Despite our deep understanding of visual signal transduction, little is known about how the outer segment is populated by proteins performing this function. Indeed, nearly all mechanistic studies of outer segment protein trafficking have been devoted to rhodopsin (*Nemet et al., 2015*; *Wang and Deretic, 2014*), which is a GPCR visual pigment comprising the majority of the outer segment membrane protein mass (*Palczewski, 2006*). The mechanisms responsible for outer segment delivery of other transmembrane proteins remain essentially unknown. Some of them contain short outer segment targeting signals, which can be identified through site-specific mutagenesis (*Deretic et al., 1998*; *Li et al., 1996*; *Pearring et al., 2014*; *Salinas et al., 2013*; *Sung et al., 1994*; *Tam et al., 2000*; *2004*). A documented exception is retinal guanylate cyclase 1 (GC-1), whose exhaustive mutagenesis did not yield a distinct outer segment targeting motif (*Karan et al., 2011*).

GC-1 is a critical component of the phototransduction machinery responsible for synthesizing the second messenger, cGMP (*Wen et al., 2014*). GC-1 is the only guanylate cyclase isoform expressed

**eLife digest** Our vision begins with light being captured by the rod and cone photoreceptor cells of the retina at the back of the eye. Photoreceptors have large antennae, termed outer segments, which contain specialized proteins that produce electrical signals when stimulated by light. However, it remains mostly unknown how these signaling proteins are delivered specifically to this part of the cell.

A light-capturing receptor called rhodopsin is by far the most abundant component of the outer segment and the only one whose transport route through the cell has been well mapped. By examining mice that had been genetically modified to lack rhodopsin, Pearring et al. have now investigated whether the rhodopsin transport pathway also delivers other signaling proteins to the outer segment.

These studies revealed that a protein called guanylate cyclase 1, which makes a messenger molecule inside rod and cone cells, is the only outer segment protein whose delivery to the outer segment and stability inside the cell rely on rhodopsin. Guanylate cyclase literally 'piggybacks' on rhodopsin on its route to the outer segment, as these proteins bind to one another . Therefore, rhodopsin is not just a light-sensing receptor protein; it also serves as a "trafficking guide" for another key protein in the same signaling pathway.

As Pearring et al. have shown that the majority of outer segment proteins are delivered independently of rhodopsin, future studies will need to search for alternative protein transport pathways in photoreceptor cells. Whether receptor molecules other than rhodopsin stabilize and deliver fellow members of their signaling pathways to specific cell structures also remains to be discovered.

in the outer segments of cones and the predominant isoform in rods (*Baehr et al., 2007*; *Yang et al., 1999*). GC-1 knockout in mice is characterized by severe degeneration of cones and abnormal light-response recovery kinetics in rods (*Yang et al., 1999*). Furthermore, a very large number of GC-1 mutations found in human patients causes one of the most severe forms of early onset retinal dystrophy, called Leber's congenital amaurosis (*Boye, 2014*; *Kitiratschky et al., 2008*). Many of these mutations are located outside the catalytic site of GC-1, which raises great interest in understanding the mechanisms of its intracellular processing and trafficking.

In this study, we demonstrate that, rather than relying on its own targeting motif, GC-1 is transported to the outer segment in a complex with rhodopsin. We conducted a comprehensive screen of outer segment protein localization in rod photoreceptors of rhodopsin knockout ($Rho^{-/-}$) mice and found that GC-1 was the only protein severely affected by this knockout. Next, we showed that this unique property of GC-1 is explained by its interaction with rhodopsin, which likely initiates in the biosynthetic membranes and supports both intracellular stability and outer segment delivery of this enzyme. These findings explain how GC-1 reaches its specific intracellular destination and also expand the role of rhodopsin in supporting normal vision by showing that it guides trafficking of another key phototransduction protein.

## Results

### GC-1 is the outer segment-resident protein severely down-regulated in rhodopsin knockout rods

This study was initiated by testing the hypothesis that rhodopsin, by far the most abundant outer segment protein whose proper transport is essential for formation of this organelle, may affect trafficking and abundance of outer segment membrane proteins. This was accomplished by a comprehensive examination of outer segment protein localization in rods of $Rho^{-/-}$ mice. Normal outer segments are cylindrical structures filled with an ordered stack of several hundred membrane discs (*Figure 1A*). In contrast, $Rho^{-/-}$ rods only develop small ciliary extensions filled with disorganized membrane material (*Figure 1A*; *Humphries et al., 1997*; *Lee et al., 2006*; *Lem et al., 1999*). Despite this morphological defect, two outer segment-specific proteins, peripherin and R9AP, have

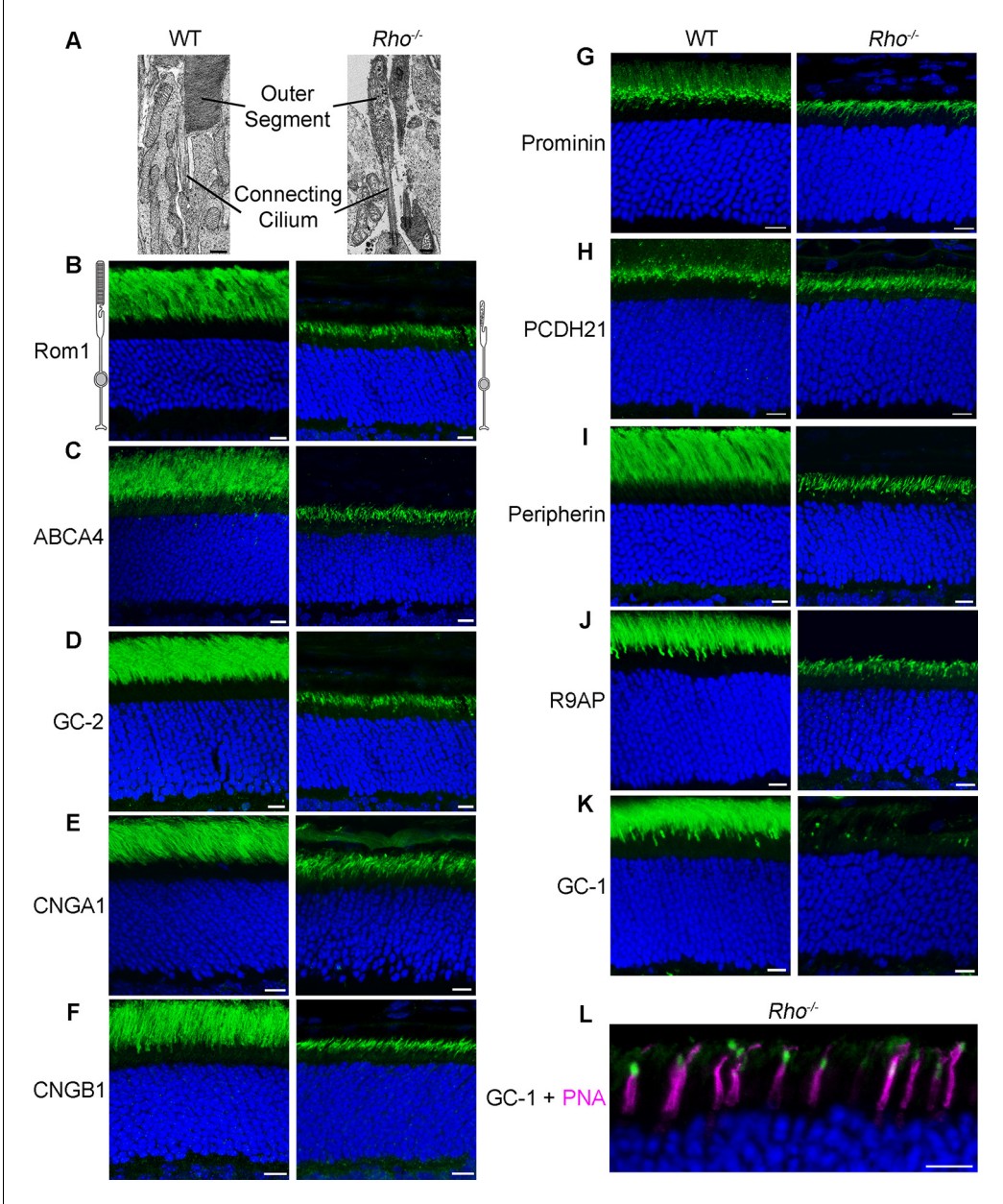

**Figure 1.** Localization of outer segment membrane proteins in wild-type (WT) and *Rho*⁻ᐟ⁻ retinas. (**A**) Electron micrographs showing the outer segment and connecting cilium in WT and *Rho*⁻ᐟ⁻ rods (scale bar 500 nm). (B–K) Immunofluorescent localization of individual outer segment proteins in WT and *Rho*⁻ᐟ⁻ retinal cross-sections: (**B**) Rom-1; (**C**) ABCA4; (**D**) guanylate cyclase 2 (GC-2); (**E**) cyclic nucleotide gated (CNG) α1; (**F**) CNGβ1; (**G**) prominin; (**H**) protocadherin 21 (PCDH21); (**I**) peripherin; (**J**) R9AP; and (**K**) GC-1. (**L**) Double labeling of GC-1 (green) and the cone maker, PNA (magenta). Here and in the following figures, the identity of antibodies used in each panel is indicated in 'Materials and methods'. Scale bars, 10 μm. Nuclei are stained by Hoechst (blue).

been previously shown to reliably target to this ciliary extension (*Lee et al., 2006*; *Pearring et al., 2014*). We broadened this analysis to include the majority of transmembrane outer segment proteins.

We analyzed ten proteins, whose antibodies have been verified in the corresponding knockout controls. Five of these proteins are components of the phototransduction cascade (R9AP, GC-1, GC-2, CNGα1, and CNGβ1), two support disc structure (peripherin and Rom1), one is a membrane lipid flippase (ATP-binding cassette transporter A4, ABCA4), and the last two are thought to participate

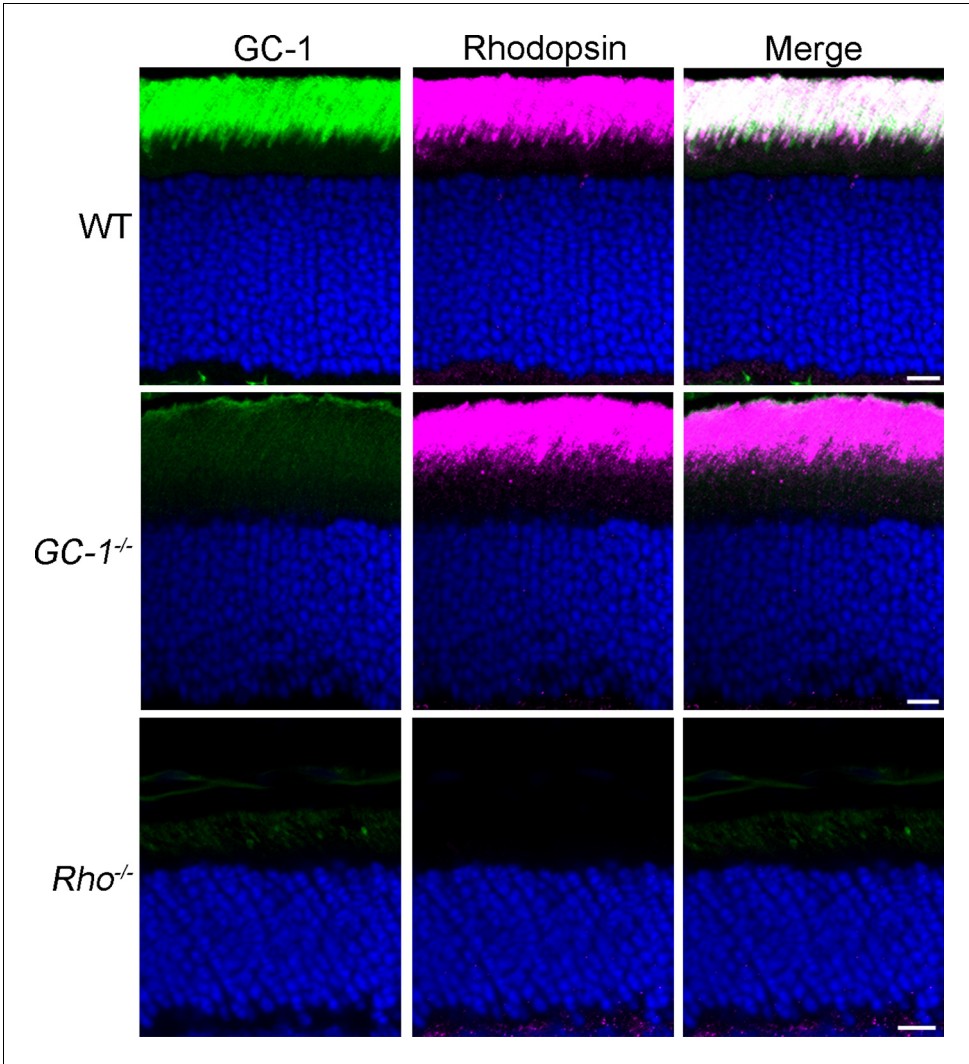

**Figure 2.** Rhodopsin expression and outer segment localization do not rely on GC-1. Rhodopsin (magenta) and GC-1 (green) were co-immunostained in retinal cross-sections from wild-type (WT), $GC-1^{-/-}$, and $Rho^{-/-}$ mice. Scale bar, 10 μm. Nuclei stained in blue.

in photoreceptor disc morphogenesis (protocadherin 21 and prominin). All experiments were performed with animals sacrificed on postnatal day 21 when the rudimentary outer segments of $Rho^{-/-}$ rods are fully formed, but photoreceptor degeneration that eventually occurs in these mice remains minimal.

Remarkably, nine out of ten proteins were localized specifically to the ciliary extensions of the $Rho^{-/-}$ rods. They included Rom1, ABCA4, GC-2, CNGα1, CNGβ1, protocadherin 21, and prominin (*Figure 1B–H*), as well as previously reported R9AP and peripherin (*Figure 1I,J*). A striking exception was GC-1, which displayed a punctate pattern in the outer segment layer with no distinct signal in rod ciliary extensions (*Figure 1K*). Further analysis using a cone marker, peanut agglutinin, revealed that the GC-1-positive puncta corresponds to cone outer segments (*Figure 1L*; note that cone outer segments in $Rho^{-/-}$ mice are smaller than normal). Faint fluorescent signal outside the cone outer segments was indistinguishable from non-specific background in the outer segment layer of GC-1 knockout mice (*Gucy2e$^{-/-}$*; *Figure 2*). Interestingly, this effect was not reciprocal. As documented in a previous study (*Baehr et al., 2007*), the knockout of GC-1 was not associated with reduction or mislocalization of rhodopsin from rod outer segments (*Figure 2*).

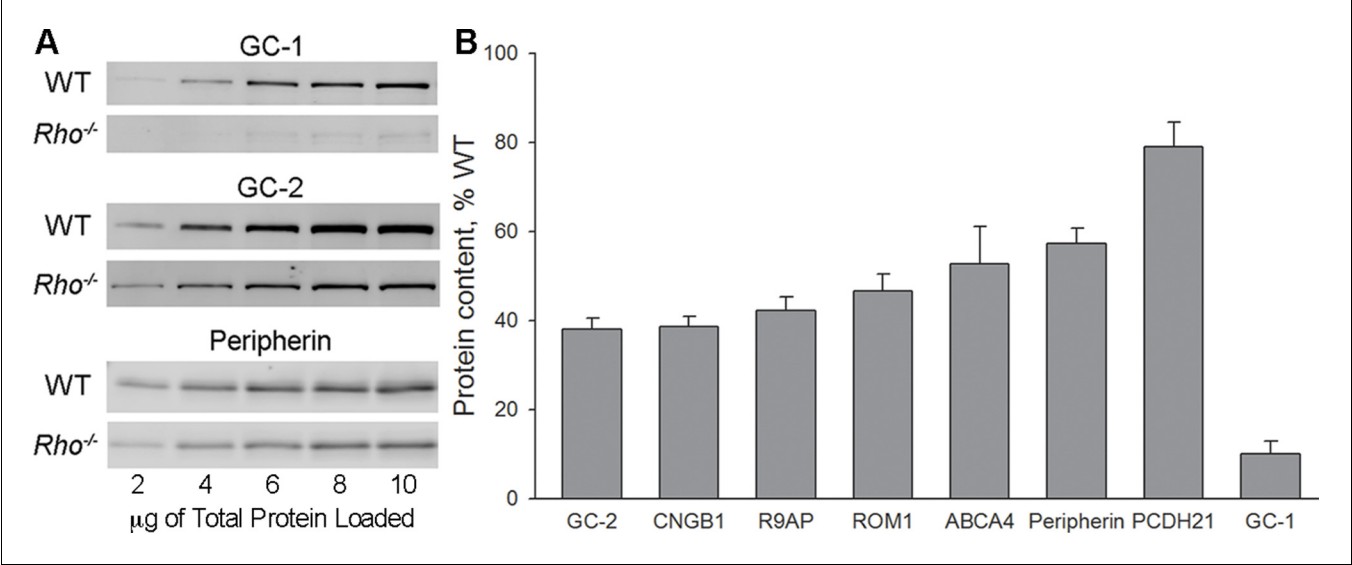

**Figure 3.** Quantification of outer segment transmembrane proteins in $Rho^{-/-}$ retinas at P21.   (**A**) Representative Western blots show serial dilutions of wild-type (WT) and $Rho^{-/-}$ retinal lysates for three proteins (guanylate cyclase 1 [GC-1], GC-2, and peripherin). The fluorescent signal produced by each band in the serial dilution was plotted and used to calculate the amount of each protein in $Rho^{-/-}$ lysate. In these examples, GC-1 was to 10% of its WT content, GC-2 to 38%, and peripherin to 57%. (**B**) Expression levels of outer segment transmembrane proteins in $Rho^{-/-}$ retinal lysates calculated as % WT. A minimum of four independent experiments was performed for each protein. Error bars represent SEM.

The following figure supplements are available for Figure 3:

**Figure supplement 1.** Transcript levels of GC-1 in the retinas of WT and $Rho^{-/-}$ mice.

We then used quantitative Western blotting to measure the amounts of outer segment proteins in the retinas of $Rho^{-/-}$ knockout mice. Availability of suitable antibodies allowed us to analyze eight of the initial ten proteins (**Figure 3**). Serial dilutions of retinal lysates from wild-type (WT) and $Rho^{-/-}$ mice were run on the same blot (such as examples in **Figure 3A**) and the relative protein amounts were calculated using WT data to generate calibration curves. We found that proteins retaining their normal outer segment localization (**Figure 1**) were all expressed at 40–80% WT levels (**Figure 3B**). Considering how small the ciliary extensions of $Rho^{-/-}$ rods are, this amount of protein expression is quite remarkable and suggests a high density of protein packing.

Once again the outlier was GC-1 whose content in $Rho^{-/-}$ retinas was only 10 ± 3% (SEM, n=5) of WT (**Figure 3A,B**). Considering that (1) a large fraction of this GC-1 is expressed in cones (**Figure 1L**); (2) cones comprise 3% of mouse photoreceptors; and (3) cones express more GC-1 than rods (**Dizhoor et al., 1994**), our most conservative estimate is that GC-1 in $Rho^{-/-}$ rods is reduced by at least 95%. This reduction is not caused by a loss of the GC-1 transcript, since qRT-PCR showed that the amount of GC-1 mRNA in $Rho^{-/-}$ retinas was actually twice higher than in WT retinas (**Figure 3—figure supplement 1**), perhaps reflecting a feedback mechanism attempting to compensate for the missing GC-1 enzyme.

Taken together, the data reported in **Figures 1 and 3** demonstrate that GC-1 is unique among outer segment transmembrane proteins in its reliance on rhodopsin for intracellular stability and outer segment localization. We next addressed the mechanistic basis for this phenomenon.

## GC-1 stability and trafficking require the transmembrane core of rhodopsin but not its outer segment targeting domain

We first asked whether the stability and localization of GC-1 in rods require rhodopsin itself or the activity of rhodopsin's trafficking pathway. The outer segment localization of rhodopsin relies on its C-terminal sequence (including the VXPX targeting motif), which is both necessary and sufficient for rhodopsin delivery (**Deretic et al., 1998**; **Li et al., 1996**; **Sung et al., 1994**; **Tam et al., 2000**). Taking advantage of the fact that adding this sequence to other proteins enables their specific outer

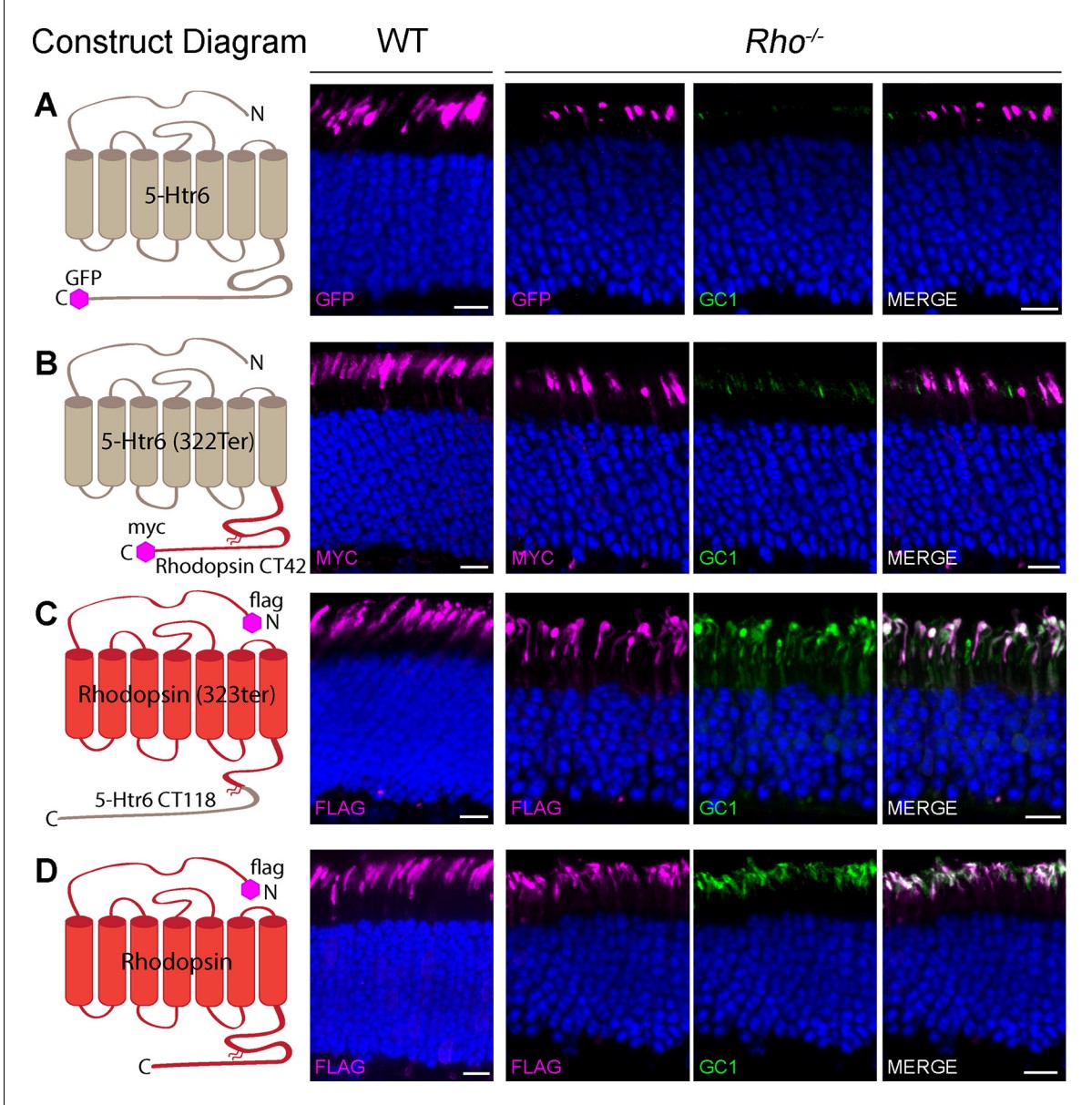

**Figure 4.** Guanylate cyclase 1 (GC-1) rescue in Rho-/- rods requires the seven-helical core structure of rhodopsin. Wild-type (WT) and *Rho*[-/-] rods were transfected with (**A**) full-length Htr6, (**B**) seven-helical Htr6 core fused to rhodopsin's C-terminus, (**C**) seven-helical rhodopsin core fused to the C-terminus of Htr6, (**D**) full-length rhodopsin. Sections from WT retinas were stained for each recombinant chimera using anti-green fluorescent protein (GFP), anti-myc, or anti-FLAG antibodies (magenta, each chimera's tag is depicted in the construct diagram). Sections from *Rho*[-/-] retinas were co-stained for GFP, myc, or FLAG (magenta, left panel) and endogenous GC-1 using anti-GC1 antibodies (green, middle panel). The merged images from *Rho*[-/-] sections are shown in the right panel. Scale bar, 10 μm. Nuclei are stained by Hoechst (blue).

segment trafficking (*Tam et al., 2000*; reviewed in *Pearring et al., 2013*), we used chimeric proteins to address this question. Two chimeras were generated by exchanging the seven-helical cores and C-terminal sequences between rhodopsin and another GPCR, serotonin receptor Htr6, whose topology is close to rhodopsin. Although this receptor is not endogenous to rods, we found that both full-length Htr6 and its rhodopsin chimeras robustly express and reliably target to rod outer segments of WT and *Rho*[-/-] mice. This is shown in *Figure 4*, where these constructs were introduced into rods by the technique of *in vivo* electroporation.

Despite its strong expression in outer segments of *Rho*[-/-] rods, Htr6 did not restore their GC-1 content (*Figure 4A*) and neither did the chimera containing the Htr6 seven-helical core fused to

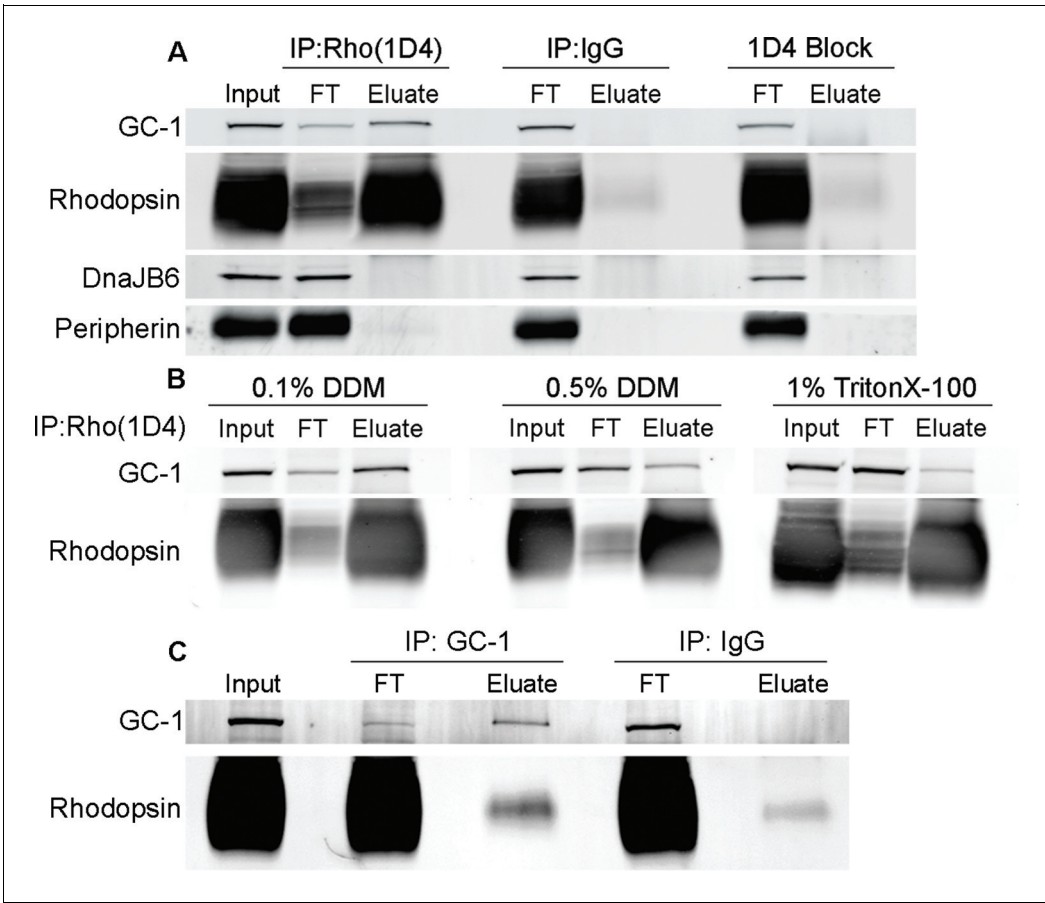

**Figure 5.** Guanylate cyclase 1 (GC-1) co-precipitation with rhodopsin from mouse retinal lysate. (**A**) GC-1 and rhodopsin co-precipitation by monoclonal anti-rhodopsin antibody 1D4. Wild-type (WT) mouse retinal lysate (Input) was incubated with 1D4 antibody and then bound to protein A/G beads. After the unbound material in flow through (FT) was removed, the beads were washed and bound proteins were eluted (Eluate) and analyzed by Western blotting for GC-1, rhodopsin, DnaJB6, and peripherin. Non-specific protein binding was probed using either non-immune mouse IgG or 1D4 antibody treated with its epitope blocking peptide. (**B**) Co-precipitation of GC-1 and rhodopsin by the 1D4 antibody from retinal membranes solubilized under different detergent conditions. (**C**) Rhodopsin and GC-1 co-precipitation by monoclonal anti-GC-1 antibody 1S4. $Rho^{+/-}$ mouse retinal lysate (Input) was incubated with 1S4 antibody bound to protein A/G beads. After the unbound material in flow through (FT) was removed, bound proteins were eluted from the beads (Eluate) and analyzed by Western blotting for GC-1 and rhodopsin. Non-specific rhodopsin binding was probed using non-immune mouse IgG. Protein loading on each lane was normalized to input in all panels.

rhodopsin's C-terminus (*Figure 4B*). In contrast, expression of the chimera containing rhodopsin's core fused to Htr6's C-terminus did restore GC-1 in the outer segments of transfected rods (*Figure 4C*), as efficiently as rhodopsin (*Figure 4D*). Therefore, both intracellular stability and outer segment delivery of GC-1 rely directly on rhodopsin and not on the activity of the ciliary trafficking pathway driven by rhodopsin's C-terminus. The most straightforward explanation of this finding is that stability and trafficking of GC-1 are being supported by its binding to rhodopsin – a hypothesis tested in the next set of experiments.

## GC-1 is a rhodopsin-interacting protein

We investigated whether GC-1 and rhodopsin interact with one another by co-precipitating them from mouse retinal membranes solubilized in a mild detergent, dodecyl maltoside. The experiment in *Figure 5A* demonstrates that a large fraction of GC-1 can be co-precipitated with rhodopsin using the monoclonal anti-rhodopsin antibody 1D4. The specificity of this co-precipitation was established

by replacing 1D4 with non-immune mouse IgG and by performing the experiment in the presence of the 1D4 epitope-blocking peptide (*Hodges et al., 1988*). Neither rhodopsin nor GC-1 was precipitated under these conditions. We also probed the 1D4 precipitate for the chaperone protein, DnaJB6 (*Figure 5A*), which was previously shown to link GC-1 to the intraflagellar transport (IFT) particle for ciliary transport (*Bhowmick et al., 2009*). Whereas a strong DnaJB6 staining was identified in the retinal lysate, it did not precipitate with the rhodopsin-GC-1 complex. This suggests that, unlike the interaction with rhodopsin, the GC-1 interaction with DnaJB6 is transient. Following a recent report that another rhodopsin-binding protein is peripherin (*Becirovic et al., 2014*), we also probed the rhodopsin precipitate for peripherin. However, no appreciable fraction of peripherin was found in precipitate, both under our experimental conditions and in membranes dissolved in Triton X-100 as in their study. Given that these authors did not show what fraction of total peripherin was precipitating with rhodopsin in their assays, it is hard to fully reconcile these observations, although it is highly unlikely that rhodopsin's interaction with peripherin could be as prominent as that with GC-1.

Importantly, the efficiency of GC-1 co-precipitation with rhodopsin was highly dependent on the type and concentration of detergent used for membrane solubilization. The co-precipitated fraction of GC-1 diminished when we increased the concentration of dodecyl maltoside, or used Triton X-100 instead (*Figure 5B*). Such instability of membrane protein complexes in detergent solutions is a common phenomenon (*Prive, 2007*), and we believe that this likely hindered identification of GC-1's interaction with rhodopsin in the past.

The reciprocal co-precipitation of rhodopsin by the monoclonal anti-GC-1 antibody 1S4 is shown in *Figure 5C*. The challenge of this experiment was that rhodopsin in mouse rods is expressed at a 1400-fold molar excess over GC-1 (*Peshenko et al., 2011b*). Therefore, the theoretical limit of rhodopsin bound to GC-1 is only 0.07% of its total amount, and its detection requires special measures to reduce non-specific rhodopsin binding to the beads. We addressed this issue by employing two strategies. First, we improved the molar ratio between GC-1 and rhodopsin by using retinal lysates from $Rho^{+/-}$ mice, which express twice less rhodopsin without affecting the amounts of other outer segment proteins (*Calvert et al., 2001*; *Liang et al., 2004*). Second, we used a minimal amount of beads fully saturated with anti-GC-1 antibodies, just sufficient to precipitate the majority of GC-1 in the lysate. Under these conditions, rhodopsin precipitation with anti-GC-1 antibody exceeded non-specific background with non-immune IgG by over six-fold (6.5 ± 1.7, n=3).

Taken together and combined with our finding that the intracellular stability of GC-1 is critically dependent on the presence of rhodopsin, these data demonstrate that rhodopsin and GC-1 form a complex in photoreceptor cells.

## Discussion

The findings reported in this study expand our understanding of how the photoreceptor's sensory cilium is populated by its specific membrane proteins. We have found that rhodopsin serves as an interacting partner and a vehicle for ciliary delivery of a key phototransduction protein, GC-1. This previously unknown function adds to the well-established roles of rhodopsin as a GPCR visual pigment and a major building block of photoreceptor membranes. We further showed that GC-1 is unique in its reliance on rhodopsin, as the other nine proteins tested in this study were expressed in significant amounts and faithfully localized to rod outer segments in the absence of rhodopsin.

Our data consolidate a number of previously published observations, including a major puzzle related to GC-1: the lack of a distinct ciliary targeting motif encoded in its sequence. The shortest recombinant fragment of GC-1 which localized specifically to the outer segment was found to be very large and contain both transmembrane and cytoplasmic domains (*Karan et al., 2011*). Our study shows that GC-1 delivery requires rhodopsin and, therefore, can rely on specific targeting information encoded in the rhodopsin molecule. Interestingly, we also found that this information can be replaced by an alternative ciliary targeting sequence from a GPCR not endogenous to photoreceptors. This suggests that the functions of binding/stabilization of GC-1 and ciliary targeting are performed by different parts of the rhodopsin molecule.

Our findings also shed new light on the report that both rhodopsin and GC-1 utilize IFT for their ciliary trafficking and co-precipitate with IFT proteins (*Bhowmick et al., 2009*). The authors hypothesized that GC-1 plays a primary role in assembling cargo for the IFT particle bound for ciliary

delivery. Our data suggest that it is rhodopsin that drives this complex, at least in photoreceptor cells where these proteins are specifically expressed. Unlike GC-1's reliance on rhodopsin for its intracellular stability or outer segment trafficking, rhodopsin does not require GC-1 as its expression level and localization remain normal in rods of GC-1 knockout mice (*Baehr et al., 2007*; and this study). The outer segment trafficking of cone opsins is not affected by the lack of GC-1 either (*Baehr et al., 2007*; *Karan et al., 2008*), although GC-1 knockout cones undergo rapid degeneration, likely because they do not express GC-2 – an enzyme with redundant function. The primary role of rhodopsin in guiding GC-1 to the outer segment is further consistent with rhodopsin directly interacting with IFT20, a mobile component of the IFT complex responsible for recruiting IFT cargo at the Golgi network (*Crouse et al., 2014*; *Keady et al., 2011*).

It was also reported that GC-1 trafficking requires participation of chaperone proteins, most importantly DnaJB6 (*Bhowmick et al., 2009*). Our data suggest that GC-1 interaction with DnaJB6 is transient, most likely in route to the outer segment, since we were not able to co-precipitate DnaJB6 with GC-1 from whole retina lysates (*Figure 5*). In contrast, the majority of GC-1 co-precipitates with rhodopsin from these same lysates, suggesting that these proteins remain in a complex after being delivered to the outer segment. Although our data do not exclude that the mature GC-1-rhodopsin complex may contain additional protein component(s), our attempts to identify such components by mass spectrometry have not yielded potential candidates.

Interestingly, GC-1 was previously shown to stably express in cell culture where it localizes to either ciliary or intracellular membranes (*Bhowmick et al., 2009*; *Peshenko et al., 2015*). This strikes at the difference between the composition of cellular components supporting membrane protein stabilization and transport in cell culture models versus functional photoreceptors. The goal of future experiments is to determine whether these protein localization patterns would be affected by co-expressing GC-1 with rhodopsin, thereby gaining further insight into the underlying intracellular trafficking mechanisms.

Finally, GC-1 trafficking was reported to depend on the small protein RD3, which is thought to stabilize both guanylate cyclase isoforms, GC-1 and GC-2, in biosynthetic membranes (*Azadi et al., 2010*; *Zulliger et al., 2015*). In the case of GC-1, this stabilization would be complementary to that by rhodopsin and potentially could take place at different stages of GC-1 maturation and trafficking in photoreceptors. Another proposed function of RD3 is to inhibit the activity of guanylate cyclase isoforms outside the outer segment in order to prevent undesirable cGMP synthesis in other cellular compartments (*Peshenko et al., 2011a*).

In summary, this study explains how GC-1 reaches its intracellular destination without containing a dedicated targeting motif, expands our understanding of the role of rhodopsin in photoreceptor biology and extends the diversity of signaling proteins found in GPCR complexes to a member of the guanylate cyclase family. Provided that the cilium is a critical site of GPCR signaling in numerous cell types (*Schou et al., 2015*), it would be interesting to learn whether other ciliary GPCRs share rhodopsin's ability to stabilize and deliver fellow members of their signaling pathways.

## Materials and methods

### Animals
WT C57BL/6J mice were from Jackson Labs (Bar Harbor, ME), WT CD-1 mice were from Charles River, *Rho* $^{-/-}$ mice (*Lem et al., 1999*) were kindly provided by Janis Lem (Tufts University), and fixed eyecups from *Gucy2e* $^{-/-}$ mice (*Yang et al., 1999*) were kindly provided by Shannon Boye (University of Florida).

### Primary antibodies
The following antibodies (with corresponding figures indicated) were generously provided by: David Garbers, University of Texas Southwestern (pAb L670, anti-GC2; *Figures 1, 3, 5*); Alexander Dizhoor, Salus University (pAb KHD, anti-RetGC1; *Figures 3, 5*); Wolfgang Baehr, University of Utah (mAb 1S4, anti-RetGC1; *Figures 1, 2, 4, 5*); Robert Molday, University of British Columbia (mAb 1D1 PMC, anti-CNGα1; *Figure 1*); Steven Pittler, University of Alabama at Birmingham (pAb, anti-CNGβ1; *Figures 1, 3*); Gabriel Travis, University of California Los Angeles (pAb, anti-peripherin residues 296–346; *Figures 1, 3*); Jeremy Nathans, Johns Hopkins University (pAb, anti-protocadherin 21

C-terminus; *Figures 1, 3*); Stefan Heller, Stanford University (pAb, anti-R9AP residues 144–223; *Figures 1, 3*).

The polyclonal antibody against Rom-1 was generated in our laboratory (*Gospe et al., 2011*) (*Figures 1, 3*). Commercial antibodies were: mAb 1D4, anti-rhodopsin (Abcam, Cambridge, MA, *Figures 1, 3, 5*); pAb, anti-rhodopsin N-terminus (Sigma, St. Louis, MO; *Figure 2*); pAb M-18, anti-ABCA4 (Santa Cruz, Dallas, TX; *Figure 3*); pAb, anti-ABCA4 C-terminus (Everest Biotech, Ramona, CA; *Figure 1*); mAb 13A4, anti-prominin (eBioscience, San Diego, CA; *Figure 1*); mAb M2, anti-FLAG (Sigma; *Figure 4*) and pAb, anti-FLAG (Pierce, Grand Island, NY; *Figure 4*); pAb, anti-GFP conjugated to Alex Fluor 488 (Molecular Probes, Grand Island, NY; *Figure 4*); pAb 71D10, anti-Myc-Tag (Cell Signaling, Danvers, MA; *Figure 4*); pAb, anti-DnaJB6 (Thermo Scientific, Grand Island, NY; *Figure 5*).

## Immunofluorescence

Posterior eyecups from C57BL/6J, *Rho* $^{-/-}$, or *GC1* $^{-/-}$ mice were fixed for 1 hr with 4% paraformaldehyde in mouse Ringer's solution, rinsed three times in Ringer's, and embedded in 4% UltraPure agarose (Invitrogen, Grand Island, NY). Cross-sections of 100 μm were collected using a vibratome (Leica Biosystems, Buffalo Grove, IL), placed in 24-well plates, and blocked in 5% goat serum and 0.5% Triton X-100 in PBS for 1 hr at 22°C. Sections were incubated with primary antibodies in blocking solution overnight at 4°C, rinsed three times, and incubated with goat secondary antibodies conjugated with Alexa Fluor 488, 568, or 647 (Invitrogen) in blocking solution for 2 hr at 22°C. To stain nuclei, 5 μg/ml Hoechst - (33342, Invitrogen) was used. To stain mouse cones, 1 μg/ml lectin peptide nucleic acid (PNA) conjugated to Alexa Fluor 488 (Molecular Probes) was used. Sections were mounted with Immu-Mount (Thermo Scientific) and cover-slipped. Images were acquired using a Nikon Eclipse 90i microscope and a C1 confocal scanner controlled by EZ-C1, version 3.10 software.

## Western blotting

Retinas from C57BL/6J or *Rho* $^{-/-}$ mice were collected at P21 and sonicated in 250 μl of 2% sodium dodecyl sulfate and 1× cOmplete protease inhibitor mixture (Roche, Indianapolis, IN) in phosphate-buffered saline (PBS). Lysates were cleared at 5,000 g for 10 min at 22°C. Total protein concentration was measured using the RC DC Protein Assay kit (Bio-Rad, Hercules, CA) and serial dilutions of each lysate were subjected to sodium dodecyl sulfate polyacrylamide gel electrophoresis (SDS-PAGE) (with samples not boiled). For most proteins, Western blotting was performed using secondary goat or donkey antibodies conjugated with Alexa Fluor 680 or 800 (Invitrogen) and bands were visualized and quantified using the Odyssey infrared imaging system (LiCor Bioscience). Bands of PCDH21 and CNGβ1 were visualized using goat or donkey secondary antibodies conjugated with horseradish peroxidase (HRP) for enhanced chemiluminescence (ECL) detection (ECL Prime, GE Healthcare, Pittsburgh, PA).

## In vivo electroporation and DNA constructs

DNA constructs were electroporated into the retinas of WT CD-1 or *Rho* $^{-/-}$ neonatal mice (*Matsuda and Cepko, 2004*), using a detailed protocol from *Gospe et al., 2011*. The pRho plasmid driving gene expression under the 2.2 kb bovine rhodopsin promoter was a gift from Connie Cepko (Addgene, Cambridge, MA; plasmid # 11156). All DNA constructs were cloned between the 5' AgeI and 3' NotI sites in pRho. Full-length mouse rhodopsin and peripherin were cloned from mouse retinal cDNA. A single N-terminal FLAG tag was added to the 5' end of rhodopsin using overlap extension PCR. The 5-Htr6 serotonin receptor was amplified from a mouse brain cDNA library (Stratagene, La Jolla, CA) and the coding sequence for eGFP was fused to its 3' end. All chimeric constructs were produced using overlap extension PCR. Rhodopsin was split at amino acid 323, 5-Htr6 was split at amino acid 322, and the C-terminal 63 amino acids from peripherin were used in the chimera with truncated rhodopsin. Each DNA plasmid (4 μg/μl) was mixed with a construct expressing soluble mCherry (2 μg/μl) for fluorescent identification of electroporated retinal patches. At least four animals yielding consistent results were analyzed for each construct.

## Protein co-immunoprecipitation

One C57BL/6J or $Rho^{+/-}$ retina was homogenized in 200 µl of PBS with 1× phosphatase inhibitor cocktail (PhosSTOP, Roche), 1× cOmplete protease inhibitor mixture (Roche), and either n-dodecyl β-D-maltoside or Triton X-100 at desired concentration. Gentle homogenization was performed using a pestle on ice without vortexing or sonication. Lysates were cleared at 100,000 g for 20 min at 4°C and 20 µl aliquots were incubated with primary antibodies overnight at 4°C under continuous rotation (5 µg of anti-rhodopsin antibody 1D4, 0.2 µg of anti-GC-1 antibody 1S4, or mouse monoclonal IgG, Santa Cruz). For epitope blocking, rhodopsin 1D4 peptide (AnaSpec, Fremont, CA) was added to lysate with 1D4 antibody at a final concentration of 2 mM. Protein A/G magnetic beads (Pierce) were incubated with the lysate under rotation for 15 min at 22°C; 25 µl of beads were used to precipitate antibodies bound to rhodopsin, while 5 µl of beads were used to precipitate GC-1. Flow through was collected and beads were washed in 100 µl of the corresponding lysate buffer before being eluted with 20 µl of 2% sodium dodecyl sulfate (SDS) in PBS. Finally, 5 µl of 6× sample buffer with 100 mM dithiothreitol (DTT) were added to each sample (input, flow through, eluate) for SDS-PAGE. Samples were not boiled.

## Quantitative RT-PCR

Retinas were collected from C57BL/6J or $Rho^{-/-}$ mice at P21, and RNA was extracted using RNeasy Mini Kit (Qiagen, Valencia, CA) and adjusted to 50 ng/µl. cDNA was then synthesized from total RNA using the QuaniTect Reverse Transcriptase kit (Qiagen). GC-1 and Thy1 (internal control) primers were designed to PCR across exons 2 and 3 (GC-1 Fwd ATCCGAGATGGGCCTAGAGT, GC-1 Rev AGCCAGTTCTTCTGCAGCTT, Thy1 Fwd GTCGCTCTCCTGCTCTCAGT, and Thy1 Rev GTTATTCTCATGGCGGCAGT). The concentrations of GC-1 and Thy1 in each sample were determined using a standard curve generated by serial dilutions of the corresponding gel-purified PCR products. Real time PCR was performed in a 10 µl reaction volume with 0.5 µl of cDNA using the iQ SYBR Green Supermix (Biorad). Thermal cycling and SYBR detection was performed on a CFX96 Real Time System (Biorad).

## Acknowledgements

We are grateful to Ms. Ying Hao for performing electron microscopy, Dr. Janis Lem for providing $Rho^{-/-}$ mice, Dr. Shannon Boye for providing fixed $GC-1^{-/-}$ mouse eyecups for immunohistochemical analysis, and Drs. David Garbers, Alexander Dizhoor, Wolfgang Baehr, Robert Molday, Steven Pittler, Gabriel Travis, Jeremy Nathans, and Stefan Heller for providing their published primary antibodies. This work was supported by the NIH Grants EY22508 (JNP), EY025732 (JNP), EY025558 (WJS), EY12859 (VYA), and EY5722 (Duke University), and the Unrestricted Grant (Duke University) and Nelson Award (VYA) from Research to Prevent Blindness.

## Additional information

### Funding

| Funder | Grant reference number | Author |
| --- | --- | --- |
| National Institutes of Health | EY22508 | Jillian N Pearring |
| Research to Prevent Blindness | Non-restricted grant | Vadim Y Arshavsky |
| Research to Prevent Blindness | Nelson Trust Award | Vadim Y Arshavsky |
| National Institutes of Health | EY025732 | Jillian N Pearring |
| National Institutes of Health | EY12859 | Vadim Y Arshavsky |
| National Institutes of Health | EY5722 | Vadim Y Arshavsky |
| National Institutes of Health | EY025558 | William J Spencer |

The funders had no role in study design, data collection and interpretation, or the decision to submit the work for publication.

## Author contributions

JNP, Conception and design, Acquisition of data, Analysis and interpretation of data, Drafting or revising the article; WJS, Conception and design, Acquisition of data, Analysis and interpretation of data, Drafting of revising the article; ECL, Acquisition of data, Analysis and interpretation of data, Drafting or revising the article; VYA, Conception and design, Analysis and interpretation of data, Drafting or revising the article

## Ethics

Animal experimentation: This study was performed in strict accordance with the recommendations in the Guide for the Care and Use of Laboratory Animals of the National Institutes of Health. All of the animals were handled according to approved institutional animal care and use committee (IACUC) protocol A011-14-01 of Duke University.

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
