## [Decision Letter]

Thank you for submitting your work entitled "Guanylate cyclase 1 relies on rhodopsin for intracellular stability and ciliary trafficking" for consideration by *eLife*. Your article has been reviewed by two expert peer reviewers, and the evaluation has been overseen by a Reviewing Editor and Gary Westbrook as the Senior Editor. The experts' comments together with the Reviewing Editor's assessment form the basis of this letter, and their reviews are included at the end. The following individuals responsible for the peer review of your submission have agreed to reveal their identity: Nikolai Artemyev (Reviewer #1) and Theodore Wensel (Reviewer #2).

The reviewers and the editor were impressed with the importance and novelty of your work. The only suggestion, and it is just that – a suggestion – is to consider looking at GC1 and rhodopsin co-localization in transfected cells. We look forward to receiving the revised version of your manuscript.

Reviewer #1:

The study by Pearring et al. is interesting and well executed. The authors investigated which of the outer segment membrane resident proteins depend on rhodopsin for proper localization using Rho KO mice. Out of the ten examined proteins, only GC1 failed to traffic to the OS in the absence rhodopsin and was severely downregulated. Moreover, GC1 was found to directly interact with rhodopsin, thus suggesting the mechanism for rhodopsin in the transport of GC1. The conclusion is that GC1 is delivered to the OS in the complex with rhodopsin seems as the most probable, but not exclusive, interpretation of the findings. Hypothetically, rhodopsin may stabilize or chaperone GC1, but then mature GC1 can traffic to the OS independent of rhodopsin. Co-immunoprecipitation of GC1 and rhodopsin does not rule out the latter possibility if the binding is induced by mild destabilization of GC1 by detergent. A few additional experiments could bolster the authors' conclusions. In transfected HEK293 cells, rhodopsin localizes to the plasma membrane even in the absence of the C-terminal targeting signal, whereas GC1 is restricted to the ER. The authors can test if co-transfection with rhodopsin will target GC1 to the plasma membrane. Also, it would be helpful to confirm the binding of rhodopsin to GC1 in the OS by means other than IP (proximity ligation assay, FRET, etc.)

Reviewer #2:

This is an interesting and well-executed study revealing that binding to rhodopsin is essential for guanylate cyclase isoform GC-1 trafficking to rod outer segments. Although this appears to be a one-of-a-kind interaction in rods, it has potential implications for disease mechanisms, and there may be analogous mechanisms in other ciliated cells. The quality of the data is excellent, with quantitative comparisons being carried out by MS and by carefully calibrated immunoblotting. The comparison of IP efficiencies in different detergents is an important procedure that should be used more often for membrane protein complexes.

---

## [Author Response]

*The reviewers and the editor were impressed with the importance and novelty of your work. The only suggestion, and it is just that – a suggestion – is to consider looking at GC1 and rhodopsin co-localization in transfected cells. We look forward to receiving the revised version of your manuscript.*Reviewer #1:

The study by Pearring et al. is interesting and well executed. The authors investigated which of the outer segment membrane resident proteins depend on rhodopsin for proper localization using Rho KO mice. Out of the ten examined proteins, only GC1 failed to traffic to the OS in the absence rhodopsin and was severely downregulated. Moreover, GC1 was found to directly interact with rhodopsin, thus suggesting the mechanism for rhodopsin in the transport of GC1. The conclusion is that GC1 is delivered to the OS in the complex with rhodopsin seems as the most probable, but not exclusive, interpretation of the findings. Hypothetically, rhodopsin may stabilize or chaperone GC1, but then mature GC1 can traffic to the OS independent of rhodopsin. Co-immunoprecipitation of GC1 and rhodopsin does not rule out the latter possibility if the binding is induced by mild destabilization of GC1 by detergent. A few additional experiments could bolster the authors' conclusions. In transfected HEK293 cells, rhodopsin localizes to the plasma membrane even in the absence of the C-terminal targeting signal, whereas GC1 is restricted to the ER. The authors can test if co-transfection with rhodopsin will target GC1 to the plasma membrane. Also, it would be helpful to confirm the binding of rhodopsin to GC1 in the OS by means other than IP (proximity ligation assay, FRET, etc.)

Reviewer #2:

This is an interesting and well-executed study revealing that binding to rhodopsin is essential for guanylate cyclase isoform GC-1 trafficking to rod outer segments. Although this appears to be a one-of-a-kind interaction in rods, it has potential implications for disease mechanisms, and there may be analogous mechanisms in other ciliated cells. The quality of the data is excellent, with quantitative comparisons being carried out by MS and by carefully calibrated immunoblotting. The comparison of IP efficiencies in different detergents is an important procedure that should be used more often for membrane protein complexes.

We appreciate the suggestion to co-express GC1 and rhodopsin in cell culture, however our laboratory is not currently conducting any cell culture studies and, starting from scratch, we are unlikely to be able to complete this task within the amount of time reasonable for a revision. It might be a better idea to address it in a separate investigation, in collaboration with one of the laboratories currently conducting this cell culture work and also performing functional assays of expressed GC1. Or, perhaps, our study will inspire them to pursue such a follow-up study, whereas our focus would remain on elucidating ciliary trafficking pathways in photoreceptors. We hope the reviewer understands; we certainly do not mean to just dismiss this thoughtful comment.

Regarding the second point of Reviewer #1, we cannot completely dismiss the hypothetic possibility that rhodopsin may stabilize or chaperone GC1, but then mature GC1 can traffic to the OS independent of rhodopsin. We think, however, that our detergent result argues that rhodopsin-GC1 complex is de-stabilized rather than stabilized by detergent, as the complex was readily breaking apart when the concentration of a very mild detergent, dodecyl maltoside, was increased from a low value (0.1%) to the levels at which this detergent is used in the majority of published studies. We addressed this concern by replacing “indicating” with “suggesting” in the paragraph outlining this logic. Regarding FRET or proximity ligation assay, we thought about these options, but unfortunately they do not apply to the conditions of outer segment membranes containing 6 mM rhodopsin. This concentration is so high that rhodopsin is always within the FRET or PLA distance to other proteins populating these membranes, regardless of whether they are in a proper complex.